# IPR-NeRF: Ownership Verification meets Neural Radiance Field

## Abstract

Neural Radiance Field (NeRF) models have gained significant attention in the computer vision community in the recent past with state-of-the-art visual quality and produced impressive demonstrations. Since then, technopreneurs have sought to leverage NeRF models into a profitable business. Therefore, NeRF models make it worth the risk of plagiarizers illegally copying, re-distributing, or misusing those models. This paper proposes a comprehensive intellectual property (IP) protection framework for the NeRF model in both black-box and white-box settings, namely IPR-NeRF. In the black-box setting, a diffusion-based solution is introduced to embed and extract the watermark via a two-stage optimization process. In the white-box setting, a designated digital signature is embedded into the weights of the NeRF model by adopting the sign loss objective. Our extensive experiments demonstrate that not only does our approach maintain the fidelity (*i.e.*, the rendering quality) of IPR-NeRF models, but it is also robust against both ambiguity and removal attacks compared to prior arts.

## 1 Introduction

Neural Radiance Field (NeRF) Mildenhall et al. (2020) is a novel view synthesis that employs volume rendering and implicit neural representation (INR) through a neural network to learn 3D scene geometry and lighting from camera-calibrated 2D images. Due to its impressive photorealistic rendering performance, NeRF models have found lucrative business opportunities such as in autonomous driving (Sucar et al., 2021; Rosinol et al., 2022) and urban mapping (Xiangli et al., 2022; Rematas et al., 2022). Down the road, NeRF models could complement other techniques for representing 3D objects in the metaverse, virtual/augmented reality, and digital twins more efficiently, accurately and realistically (Hong et al., 2022; Zhao et al., 2022). Thus, the protection of Intellectual Property Rights (IPR) of the NeRF models requires immediate attention as it is worth the risk of plagiarizers illegally copying, re-distributing, or misusing those models for financial gain or personal interests.

| Method | Training Image | Embedded Watermark | Extracted Watermark |
|---|---|---|---|
| DeepStega Baluja (2020) | | | |
| HiDDen Zhu et al. (2018) | | | |

Figure 1: Preliminary results of conventional 2D steganography methods resulted in an unrecoverable watermark during NeRF rendering.

From the literature, a comprehensive IPR protection framework for NeRF models is yet to be established. It is important to note that the current protection schemes primarily target Convolutional Neural Networks (CNNs) (Uchida et al., 2017; Zhang et al., 2020; Fan et al., 2022), Generative Adversarial Networks (GANs) (Ong et al., 2021), and Recurrent Neural Networks (RNNs) (Tan et al., 2022; Rathi et al., 2022; He et al., 2022). Nonetheless, there exist multiple challenges when designing a protection framework for NeRF models: **(a)** Existing black-box protection schemes (*e.g.*, (Merrer et al., 2017; Adi et al., 2018; Quan et al., 2021)) *cannot be directly applied* to NeRF due to the distinct input-output behaviour of NeRF models (*i.e.*, coordinates and conditional information as input and RGB color as

Figure 2: **(Left)** Overview of our proposed IPR protection framework (namely IPR-NeRF) in both white and black box settings. The concept is to seamlessly embed a signature and a watermark into a standard NeRF model without deteriorating its rendering performance. During ownership verification, we extract the embedded signature and watermark for ownership claims. **(Right)** We present a comparison of rendering quality between the original NeRF and our proposed IPR-NeRF.

output). **(b)** Simply watermarking the rendered samples with existing deep steganography methods (*e.g.*, HiDDeN (Zhu et al., 2018) and DeepStega (Baluja, 2020)) either having *low rendering quality* or the watermark *cannot be robustly extracted* (see Fig. 1).

This paper introduces IPR-NeRF, a comprehensive IPR protection framework explicitly designed for NeRF models as depicted in Fig. 2. For black-box ownership protection, we start by embedding a designated watermark into the rendered scene by NeRF with jointly optimize a diffusion-based watermark detector network to ensure precise watermark extraction from NeRF rendering. The diffusion-based approach enhances the robustness of extraction from varying camera poses even in cases of severe distortion (*e.g.*, noise and compression), owing to its inherent resilience to noise. For white-box ownership protection, we adopted a sign-loss objective (Fan et al., 2022) to incorporate a designated digital signature into the NeRF model's weights. This technique has demonstrated resilience against both ambiguity and removal attacks.

Our contributions can be summarized as follows:

(i) This paper is among the pioneers in investigating IPR protection for NeRF, introducing a comprehensive framework to protect NeRF in both black-box and white-box settings, aiming to prevent misuse by unauthorized parties.

(ii) We propose a diffusion-based watermark detector to effectively extract watermarks from the rendered images for black-box ownership protection (see Sec. 3.1 and Sec. 4.3).

(iii) Empirical results show that IPR-NeRF is robust against ambiguity and removal attacks, thereby establishing ownership of the NeRF model. Notably, our method maintains high fidelity compared to the original NeRF model, ensuring that rendering performance is not compromised (see Sec. 4.2). Consequently, unauthorized use of the protected NeRF model by illegal parties can be effectively prevented (see Sec. 4.4 and Sec. 4.5).

## 2 RELATED WORK

**Ownership Protection and Verification.** One of the pioneer works on *white-box* protection of the CNN model watermarking was to impose additional regularization terms on the weights parameters (Uchida et al., 2017). Nevertheless, this approach is constrained in that the internal model parameters must be accessed to extract and verify the embedded watermark. In doing so, a method for protecting the DNN model in a *black-box* setting has been proposed (Quan et al., 2021; Merrer et al., 2017; Adi et al., 2018). This involves remotely verifying ownership through API calls by embedding watermarks within the classification labels of adversarial examples present in a trigger set. Furthermore, the protection scheme combines both black-box and white-box settings which are designed to effectively withstand a range of potential attacks (Chen et al., 2019; Darvish Rouhani et al., 2019b;a; Guo & Potkonjak, 2018). Lately, there has been a surge in the proposal of passport-based verification schemes as white-box protection, aimed at enhancing robustness against ambiguity attacks (Zhang et al., 2020; Fan et al., 2022). In addition to the protection framework for CNNs, comprehensive black-box, and white-box protection frameworks have also been introduced for GANs (Ong et al., 2021) and RNNs (Tan et al., 2022; Rathi et al., 2022; Lim et al., 2022; He et al., 2022). It's important to note that the existing works discussed so far have primarily focused on watermarking techniques

for CNNs, GANs, RNNs, and 2D images instead of the NeRF model. In this work, we propose a comprehensive framework to protect NeRF models that not only maintains high fidelity, but also robust against ambiguity and removal attacks.

**Watermarking by Steganography.** An additional realm of embedding watermarks within images, referred to as image steganography, has found widespread application in protecting the ownership of digital media, particularly on 2D images (Cox et al., 2007; Wang et al., 2023; Subramanian et al., 2021). Traditional steganography methods involve the incorporation of watermarks by altering the host images, which often leads to partial distortion and a lack of robustness, even with minor modifications to the host images (Tamimi et al., 2013; Pan et al., 2011). To enhance robustness, a deep learning-based approach to steganography has been introduced. This involves the utilization of an encoder to produce a watermarked host image and a decoder to subsequently extract the embedded watermark (Baluja, 2017; 2020; Zhu et al., 2018; Hayes & Danezis, 2017; Tang et al., 2017; Zhang et al., 2019; Tancik et al., 2020). While applying a similar 2D steganography approach to watermarking the NeRF model might appear reasonable, such a strategy proves ineffective in the NeRF context. This is attributed to the embedded watermark becoming smoothed out during the rendering of the NeRF model, as demonstrated in Fig. 1 of our preliminary result. Furthermore, NeRF's representation incorporates the concept of INR, whereas the 2D images merely constitute the final output derived through NeRF's rendering process. As a result, it is desired to design a specific protection framework for NeRF models that can maintain fidelity while being robust against various attacks.

**Watermarking for INR/NeRF.** To the best of our knowledge, two recent works are closely related to ours – StegaNeRF (Li et al., 2023) and CopyRNeRF (Luo et al., 2023). StegaNeRF uses a classifier and a decoder to detect and reconstruct the watermarks that were embedded through fine-tuning a well-trained NeRF. However, this affects the rendering quality. Hence, they have to use an adaptive gradient masking method so that more significant weights in the NeRF model are masked out to minimize the impact on the rendering quality. Similarly, CopyRNeRF also has a message decoder to extract $M$-bits message from rendered views which were embedded through a message feature field as part of the NeRF model. Inspired by diffusion models (Ho et al., 2020; Song et al., 2021), our IPR-NeRF also includes a detector trained through the diffusion process because of the gradual introduction of noise during the diffusion process. This allows our IPR-NeRFnot only robust against challenges such as noisy image transformations (*e.g.*, Gaussian noise and JPEG compression) and removal attacks (see Sec. 4.5) compared to StegaNeRF but also to embed high-dimensional images instead of low-dimension message bits compared to CopyRNeRF.

## 3    IPR-NERF

This paper proposes a complete and effective ownership protection scheme for the NeRF model, catering to both black-box and white-box settings as illustrated in Fig. 2. It consists of two stages where, the former being the original NeRF training procedure. It involves the standard photometric loss between the ground truth and the rendered pixels. We denote this trained model as $\theta_0$. This is because we aim to embed our watermark into our typically trained but unmarked private/local[1] NeRF models. In the latter, a designated watermark $w$ and signature $s$ are embedded into the NeRF model through the proposed diffusion-based method described in Sec. 3.1 - Sec. 3.2. With this, we will obtain a marked NeRF model $\theta_m$ and a detector $d$ for ownership verification. The overall training steps are depicted in Algorithm 1 in Appendix.

For verification purpose[2], we recommend one can first run black-box ownership verification to obtain evidence (*i.e.*, the extracted watermark $\hat{w} = d(\theta_m)$ matches the designated watermark $w$); and then follow by initiating a trial[3] to inspect the inner weights of the NeRF model. By examining the sign of the scales of all the normalization layers in $\phi_m$, we obtain a signature $\hat{s}$. If $\hat{s}$ matches $s$, we can fully claim the ownership of the NeRF model.

---

[1]Note that once an unprotected NeRF model is released to the public, everyone may use the same technique to embed watermarks. As a result, this leads to ambiguity problems as everyone can hypocritically claim ownership of this public model.

[2]When we suspect an individual or organization posted their NeRF model without crediting the creator

[3]This is because black-box verification method may be prone to ambiguity attack (Fan et al., 2022), we need more substantial evidence for proof.

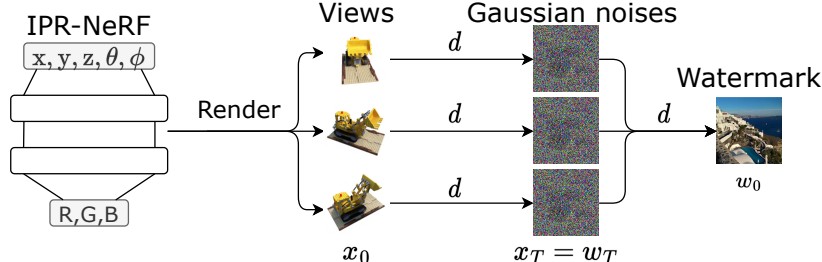

Figure 3: To retrieve the watermark from the IPR-NeRF model, our process begins by rendering various views from distinct viewpoints, resulting in the acquisition of $x_0$. Subsequently, we apply a deterministic forward diffusion process to generate noisy samples $x_T$ using the trained diffusion-based detector $d$. For each $x_T$, equivalent to $w_T$, we iteratively denoise the noisy samples through $d$ until we recover the watermark $w_0$.

### 3.1 BLACK-BOX PROTECTION SCHEME

Let $w \in \mathbb{R}^{H \times W}$ as the watermark image to be embedded *implicitly* into a NeRF model $\theta_0$ where $H$ and $W$ are the height and width of rendered views. We can recover $w = d(\theta_m(P))$ from any rendered views given any camera pose $P \in \mathcal{P}$ through our proposed detector $d$ and $\mathcal{P}$ all possible camera poses. At the end of training, the detector $d$ and the watermarked NeRF model $\theta_m$ are obtained. Inspired by (Ho et al., 2020), we propose a diffusion-based method to learn the detector $d$. This is because the nature of diffusion models to gradually add noise to the image provides natural robustness against various forms of image degradation attacks as the main objective of diffusion models is denoising. Thanks to the powerful denoising ability of diffusion models, our idea is to first convert the views $x$ into Gaussian noises, then we denoise from these Gaussian noises into the watermark $w$. As a result, we can seamlessly translate between $x$ and $w$ through $d$.

**Diffusion-based Detector.** To train the detector $d$, we first diffuse any rendered views $x = \theta_m(P) \in \mathbb{R}^{H \times W}, P \in \mathcal{P}$ into a Gaussian distribution $\mathcal{N}$. Then we reverse this process from the Gaussian distribution into the designated watermark image $w$. It can be formulated as:

$$q(x_{1:T}|x_0) := \prod_{t=1}^{T} q(x_t|x_{t-1}), \quad q(x_t|x_{t-1}) := \mathcal{N}(x_t; \sqrt{\alpha_t}x_{t-1}, (1-\alpha_t)I), \quad (1)$$

$$p_d(w_{0:T}) := q(x_T) \prod_{t=1}^{T} p_d(w_{t-1}|w_t), \quad p_d(w_{t-1}|w_t) := \mathcal{N}(w_{t-1}; \mu_d(w_t, t), (1-\alpha_t)I) \quad (2)$$

where $q$ is the diffusion process that is fixed to a Markov chain that gradually adds Gaussian noise to the rendered views $x$ according to a variance schedule $\alpha_1, \cdots, \alpha_T$ (Ho et al., 2020), $p_d$ is the reverse process that aims to denoise the Gaussian noise at any timestep $t$, and the denoiser $d$ is a learnable component that corresponds to our desired watermark detector. At timestep $t = 0$, the watermark $\hat{w} = w_0$ is extracted.

This diffusion process can be modeled by minimizing the following denoising objectives:

$$\mathcal{L}_d(d, x, w) = \mathbb{E}_{t, \{\varepsilon_x, \varepsilon_w\} \sim \mathcal{N}(0, I), P \sim \mathcal{P}} \bigg[ ||d(\sqrt{\bar{\alpha}_t}x_0 + \sqrt{1-\bar{\alpha}_t}\varepsilon_x, c_x) - \varepsilon_x||^2$$

$$+ ||d(\sqrt{\bar{\alpha}_t}w_0 + \sqrt{1-\bar{\alpha}_t}\varepsilon_w, c_w) - \varepsilon_w||^2 \bigg], \quad (3)$$

where $\bar{\alpha}_t = \prod_{i=1}^{t} \alpha_i$, $c_x, c_w$ are conditional information for $d$ to denoise for the rendered views $x$ or the watermark image $w$. Since $q(x_T)$ is sampled from a Gaussian distribution, $\mathcal{N}(0, I)$, which is the same as $q(w_T)$, the denoiser $d$ is to denoise from $\mathcal{N}(0, I)$ to respective image conditioned on $c$. As a result, we can diffuse any rendered views into Gaussian noises, and then reverse the diffusion into the watermark image.

**Watermark Extraction.** Instead of simply diffusing the rendered views into a Gaussian noise (*i.e.*, Eq. 1), we compute it by a deterministic forward process (Song et al., 2021; Preechakul et al., 2022)

given any rendered views $x$:

$$x_{t+1} = \sqrt{\alpha_{t+1}}\left(\frac{x_t - \sqrt{1-\alpha_t}\,d(x_t, t, c_x)}{\sqrt{\alpha_t}}\right) + \sqrt{1-\alpha_{t+1}}\,d(x_t, t, c_x). \tag{4}$$

At the final timestep $T$, we obtain a noisy $x_T$ which we denote as $\hat{w}_T$. Then, we can extract the watermark $\hat{w}_0$ from the computed noisy $\hat{w}_T$ by reversing deterministic diffusion as:

$$\hat{w}_{t-1} = \sqrt{\alpha_{t-1}}\left(\frac{\hat{w}_t - \sqrt{1-\alpha_t}\,d(\hat{w}_t, t, c_w)}{\sqrt{\alpha_t}}\right) + \sqrt{1-\alpha_{t-1}}\,d(\hat{w}_t, t, c_w). \tag{5}$$

At timestep $t = 0$, we obtain the extracted watermark $\hat{w} = \hat{w}_0$ and are ready to perform ownership verification. See Fig. 3 for illustration.

**Remark.** As the detector $d$ is overfitted to the distribution of $x$, it may not effectively denoise the deterministic Gaussian noises from Eq. 1 in other random images (*e.g.*, rendered views from an unprotected NeRF model). Our experiments in Sec. 4.3 reveal that we cannot obtain the correct Gaussian distribution by these alternative random images, thus preventing them from extracting valid watermark images. This also effectively prevents false positive detection.

## 3.2 WHITE-BOX PROTECTION SCHEME

We further enhance the protection of a NeRF model by embedding a designated signature $s$ into it. We adopted the sign loss objective (Fan et al., 2022) to modify the scales of all normalization layers in the NeRF model $\theta_m$. To achieve this, normalization layers were appended to the MLP implementation. The signature is encoded as ASCII codes, and subsequently translated into binary format. For the scales of the all normalization layers $\gamma_i^l \in \mathbb{R}, i \in \{1, \cdots, D\}, l \in \{1, \cdots, L\}$ where $i$ and $l$ represent $i$-th dimension and $l$-th layer, and $D$ and $L$ is the total number of dimensions and layers respectively. The modified sign loss objective is formulated as follows:

$$\mathcal{L}_s = \sum_{l=1}^{L}\sum_{i=1}^{D} max(\lambda_s - \gamma_i^l s_i^l, 0) \tag{6}$$

where $\lambda_s = 0.1$ is to prevent the magnitude of $\gamma$ from falling below $\lambda$ and avoiding a value of zero. Given a standard NeRF configuration encompassing $L = 8$ layers, with each layer comprising $D = 256$ neurons, the model can accommodate 256 ASCII characters (2048 bits).

## 3.3 LEARNING OBJECTIVE

**Photometric Error.** To minimize the impact of the watermark embedding process (*i.e.*, to prevent overfitting on $\mathcal{L}_d$ and $\mathcal{L}_s$) on the rendering quality, we employ a photometric loss as regularization:

$$\mathcal{L}_{\text{photometric}} = \mathbb{E}_{P\sim\mathcal{P}}||\theta_0(P) - \theta_m(P)||^2 \tag{7}$$

The total loss, $\mathcal{L}$, at the second-stage optimization is formulated as:

$$\mathcal{L} = \mathcal{L}_{photometric} + \lambda_d \mathcal{L}_d + \mathcal{L}_s \tag{8}$$

where $\lambda_d$ is a hyperparameter that scales the denoising objective, ensuring a balanced optimization of rendering and watermark quality within the loss function.

## 3.4 OWNERSHIP VERIFICATION

To verify a NeRF model in a black-box setting, this process involves the remote access of suspected online NeRF models by the owner via API calls for evidence collection (Li et al., 2023). With the extracted $\hat{w}$ and our watermark $w$, we employ the Structural Similarity Index (SSIM) with $w_{\text{SSIM}} = ssim(\hat{w}, w)$ to quantify the similarity between both extracted and ground-truth watermarks. To enhance reliability, a threshold of 0.75 is set (as the visibility is still evident, see Fig. 13 in Appendix). If $w_{\text{SSIM}} > 0.75$, we can claim the ownership of the NeRF model.

Meanwhile, to verify a NeRF model in a white-box setting, there are two possible scenarios: one is we can directly obtain the suspected model (*e.g.*, post the whole model online), or another one is to get enough evidence from black-box verification and initiate law enforcement to access the suspected model. Either way, we can inspect the sign of the scales and convert them into binary bits. By calculating the bit error rate (BER), we can claim ownership *iff* BER $\approx 0\%$.

|  |  | NeRF (baseline) | IPR-NeRF w/o $s$ | IPR-NeRF (Full) |
|---|---|---|---|---|
| NeRF-Synthetic | PSNR ↑ | 31.21 (±0.28) | 30.95 (±0.51) | 31.05 (±0.46) |
|  | SSIM ↑ | 0.9593 (±0.0048) | 0.9541 (±0.0083) | 0.9571 (±0.0062) |
|  | LPIPS ↓ | 0.0528 (±0.0029) | 0.0583 (±0.0037) | 0.0573 (±0.0051) |
| LLFF-Forward | PSNR ↑ | 27.59 (±0.21) | 27.13 (±0.18) | 27.39 (±0.0032) |
|  | SSIM ↑ | 0.8293 (±0.0083) | 0.8216 (±0.0105) | 0.8237 (±0.0077) |
|  | LPIPS ↓ | 0.1685 (±0.0025) | 0.1731 (±0.0027) | 0.1705 (±0.0042) |

Table 1: Fidelity: Comparison of NeRF rendering quality in different protection settings and metrics.

## 4 EXPERIMENTAL RESULTS

This section presents the empirical analysis of the proposed IPR-NeRF framework in terms of fidelity, effectiveness, and robustness against *unprotected* standard NeRF (Mildenhall et al., 2020), StegaNeRF (Li et al., 2023), and CopyRNeRF (Luo et al., 2023). While CopyRNeRF was initially designed for text embedding, we have integrated it into this study for image embedding to facilitate a more comprehensive analysis and comparison.

### 4.1 EXPERIMENTAL SETUP

**Dataset.** We employ the LLFF and NeRF-Synthetic datasets, following the original NeRF paper (Mildenhall et al., 2020). Four distinct scenes were chosen from LLFF with forward scenes (namely, **LLFF-Forward**): *Fern*, *Fortress*, *Room*, and *Flower*; while three distinct scenes were chosen from **NeRF-Synthetic** with 360° scenes: *Lego*, *Chair*, and *Drums*. The sample designated watermark used are shown in Fig. 4. In practice, one can use any unique watermark image, such as a company logo, to claim ownership and mitigate potential ambiguities.

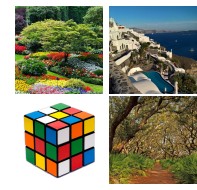

Figure 4: Samples of designated watermark employed in this paper.

**Training and Hyperparameter.** To train the NeRF in the first stage, we follow the original implementation in (Mildenhall et al., 2020) for all hyperparameters (*e.g.*, number of layers of the MLP, and positional embeddings). To train the diffusion-based detector in the second stage, we follow DDPM (Ho et al., 2020) for all hyperparameters (*e.g.*, the variance schedule, and number of timesteps). Unless mentioned explicitly, $\lambda_d = 1.0$ is used throughout our experiments and the default number of training epochs is 1000 for each scene.

**Evaluation metrics.** To measure the fidelity of a NeRF model, the Peak signal-to-noise ratio (PSNR), Structural Similarity Index (SSIM) and Learned Perceptual Image Patch Similarity (LPIPS) (Zhang et al., 2018) are used. To quantify the similarity between both extracted and ground-truth watermarks during black-box verification, we measure the SSIM and denote it as $w_{\text{SSIM}}$. Finally, we measure the bit error rate (BER) as the accuracy of the extracted signature during white-box verification. For all metrics except BER, we randomly sample 100 different camera poses to render each scene within each dataset with 5 trials and report the average score.

### 4.2 FIDELITY

This section assesses the fidelity of the proposed method in comparison to the original NeRF model, where IPR-NeRF w/o $s$ indicates embed with watermark only, and IPR-NeRF (Full) indicates embed with both watermark + signature. Table 1 reveals the following key observations: (a) The protected NeRF model closely mirrors the overall performance of the original NeRF model with a minimal deviation across all metrics. See examples in Table 2. (b) Incorporating a white-box protection scheme (signature) is deemed safe, as it exerts a negligible impact on the fidelity scores. Consequently, we can affirm that IPR-NeRF effectively preserves the fidelity of NeRF models.

### 4.3 OWNERSHIP VERIFICATION

**Black-box.** In this setting, the ownership of the NeRF model can be verified by assessing the similarity between the extracted watermark image and the ground truth watermark im-

| | | | | | | | | |
|---|---|---|---|---|---|---|---|---|
| PSNR ↑
SSIM ↑
LPIPS ↓ | 31.08
0.9568
0.0572 | 0.9631 | 31.45
0.9507
0.0583 | 0.9583 | 30.97
0.9527
0.0580 | 0.9551 | 31.36
0.9531
0.0585 | 0.9604 |
| PSNR ↑
SSIM ↑
LPIPS ↓ | 27.47
0.8394
0.1692 | 0.9762 | 27.28
0.8284
0.1706 | 0.9737 | 27.58
0.8399
0.1673 | 0.9693 | 27.24
0.8261
0.1704 | 0.9753 |

Table 2: Quantitative and qualitative results from our proposed IPR-NeRF framework involve rendering with various camera poses (left) and embedded watermarks (right), along with the corresponding extracted watermarks from NeRF-Synthetic (*Lego*) and LLFF-Forward (*Flower*) datasets.

age using the diffusion-based detector on the rendered scene of the protected NeRF model. Table 3 illustrates that, across both datasets, our proposed IPR-NeRF framework consistently achieves SSIM scores exceeding 0.95 for the extracted watermark images. This indicates a substantial similarity between the extracted watermark image and the ground truth watermark image, a similar result was also observed in StegaNeRF and CopyRNeRF. Consequently, this high SSIM score is early solid evidence to establish ownership claims and promptly identify the suspect model. See Table 2 and Table 11 in the Appendix for more visual results.

**White-box.** In this setting, the NeRF model ownership can be further verified by accessing the weights within the normalization layers and converting them into ASCII code to extract the embedded signature, as demonstrated in Table 7 in Appendix. In practical scenarios, it is recommended to embed meaningful information like the owner's name as a signature to prevent

| | | IPR-NeRF (Ours) | StegaNeRF | CopyRNeRF |
|---|---|---|---|---|
| NeRF-Synthetic | $w_{SSIM}$ ↑
BER ↓ | 0.9653
0 | 0.9671
✗ | 0.9531
✗ |
| LLFF-Forward | $w_{SSIM}$ ↑
BER ↓ | 0.9781
0 | 0.9735
✗ | 0.9692
✗ |

Table 3: Ownership Verification: Comparison of the quality of the extracted watermark, and BER between our proposed and prior arts.

ambiguity. As observed in Table 3, the extracted signature from our proposed IPR-NeRF framework achieves a BER of 0 in both datasets, demonstrating a 100% accuracy match with the embedded signature to claim ownership.

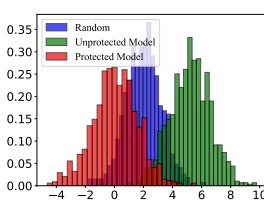

Figure 5: Histogram of $x_T$ for IPR-NeRF protected model (red), unprotected model $y_T$ (green), and random images $z_T$ (blue). x-axis represents the value and y-axis represents the normalized frequency.

**False positive detection prevention on unprotected standard NeRF model & any random images (see Appendix for experiment details).** This section aims to validate that our detector avoids detecting a watermark (false positive) on an unprotected model (*i.e.* doesn't contain an embedded watermark - a model owned by someone else). Quantitatively, the SSIM for extracted watermarks from any alternative images, when compared to the ground-truth watermark, registers only approximately 0.05-0.10. Consequently, the false positive and false negative rates, utilizing the proposed decision boundary ($w_{SSIM} > 0.75$), yield a value of 0. To understand why false positives are absent, we examine histograms from 100 views of each scene for $x_T/y_T$ and 100 random images of $z_T$. Then, we calculate the Maximum Mean Discrepancy (MMD) and Wasserstein distance (WD) between $x_T$ and $y_T$, and $z_T$. In Fig. 5, the distinct differences in the distributions of $y_T$ and $z_T$ from $x_T$ are evident, with MMD/WD discrepancies of 0.2979/2.77 and 0.8238/10.16, respectively. In summary, our detector exclusively extracts watermarks from the deterministic forward distribution of the protected model.

## 4.4 RESILIENCE AGAINST AMBIGUITY ATTACKS

This section assesses the robustness of IPR-NeRF against attempts by illegal parties to distort the embedded watermark and digital signature. These attempts may include image degradation in NeRF-rendered scenes and forged signatures, aiming to create an ambiguous situation.

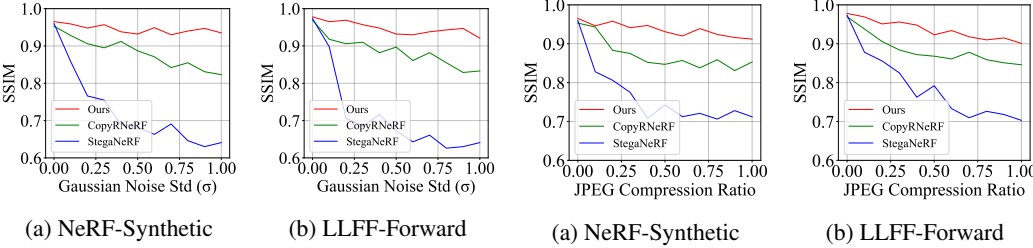

(a) NeRF-Synthetic     (b) LLFF-Forward        (a) NeRF-Synthetic     (b) LLFF-Forward

Figure 6: Robustness against Gaussian Noise.    Figure 7: Robustness against JPEG Compression.

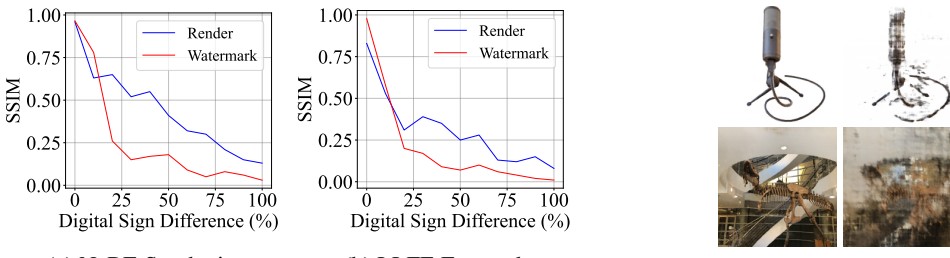

(a) NeRF-Synthetic       (b) LLFF-Forward

Figure 8: Ambiguity Attack: Quality of the rendering and extracted watermark for varying sign changes in IPR-NeRF.

Figure 9: Sample image pairs rendered by IPR-NeRF with sign toggled at 0% (left) and 10% (right).

**Image Degradation.** Here, we simulate a scenario where illegal parties attempt to distort the rendered output of the protected NeRF model. Their goal is to prevent the embedded watermark from being successfully extracted by a trained watermark detector, by degrading the rendered scene image, resulting in an ambiguous situation. Doing so demonstrates that the embedded watermark can be effectively extracted from the degraded rendered scene image.

As illustrated in both Figs. 6 - 7, our proposed diffusion-based watermark detector demonstrates better resilience against degradation attacks like Gaussian noise and JPEG compression. It consistently preserves a high-quality extracted watermark image, achieving an SSIM exceeding 0.9. In contrast, the autoencoder-based watermark detector proposed in StegaNeRF deteriorates to below SSIM = 0.7 under conditions where Gaussian noise and JPEG compression ratio are both set to 1. Although CopyRNerf performs well against image degradation attacks compared to StegaNeRF, attributed to including a distortion layer in their study, it still falls short compared to our approach. Particularly, CopyRNeRF only achieves an extracted watermark quality with an approximate 0.85. This resilience arises from the inherent ability of the proposed diffusion-based watermark detector to withstand various forms of degradation. In short, this shows that our proposed IPR-NeRF is robust against a variety of degradation attacks as compared to StegaNeRF (Li et al., 2023) and CopyRNeRF (Luo et al., 2023) watermark detection method.

**Forged Signature.** Herein, we simulate a scenario where the embedded signature was completely exposed to the attacker. The attacker's objective is to manipulate this signature by randomly toggle its sign, thereby creating an ambiguous situation. Here, we demonstrate that altering the digital signature proves challenging without sacrificing the model's overall performance.

As observed in Fig. 8, the rendering quality of IPR-NeRF significantly decreases across both datasets, despite only 10% of the sign being modified. Although the quality of the extracted watermark image substantially decreases, the quality of the rendered scene also badly deteriorates as shown qualitatively in Fig. 9. Doing so, the compromised model is essentially unusable for consumers. In short, we conclude that the signs enforced this way remain persistent against ambiguity attacks. Consequently, the illegal parties could not employ the model with a modified digital signature without compromising the rendering quality of the protected NeRF model.

## 4.5 ROBUSTNESS AGAINST REMOVAL ATTACKS

In this section, we evaluate the robustness of our proposed method in protecting the NeRF model to defend against attempts by illegal parties to remove the embedded watermark and digital signature using common model modification techniques like pruning, and overwriting.

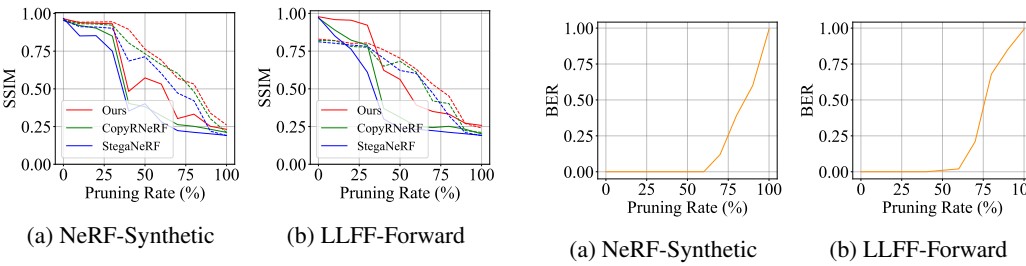

(a) NeRF-Synthetic    (b) LLFF-Forward              (a) NeRF-Synthetic    (b) LLFF-Forward

Figure 10: Removal Attack (model prun-
ing): Comparison of visual quality of rendering
(dashed line) and designated watermark (solid
line) against different pruning rates/methods.

Figure 11: Removal Attack (model pruning):
BER of signature extraction against different
pruning rates in IPR-NeRF.

**Model Pruning.** In this section, we simulate a scenario where the attacker attempts to remove the
embedded watermark and digital signature from the protected NeRF model using a model pruning
technique. We employed a global unstructured L1 pruning technique with varying pruning rates.

As illustrated in Fig. 10, up to a model pruning rate of 30%, both the rendering quality of our ap-
proach and the extracted watermark image remain preserved across both datasets. However, beyond
a pruning rate of 30%, we observed a gradual reduction in both rendering quality and watermark
integrity. Note that a detector with low SSIM means it could not detect and extract the designated
watermarks, which is equivalent to a useless model. Moreover, Fig. 11 reveals that the accuracy of
the extracted digital signature from our approach remains unaffected, exhibiting a BER of 0 even at
a model pruning rate of 60%. In summary, our findings underscore the substantial impact of model
pruning on the overall performance of the protected NeRF model, particularly on the quality of the
extracted watermark way before the embedded digital signature can be removed. Consequently, our
approach demonstrates robustness against model pruning.

**Overwriting.** In this section, we simulate a scenario where the attacker possesses comprehensive
knowledge of the NeRF model protection methods outlined in Sec. 3 of this study. The objective of
the attacker is to overwrite the existing watermark and digital signature, and subsequently replace
them with a new watermark $w'$ and digital signature $s'$ within the protected NeRF model following
the training steps as shown in Algorithm 1 (see Appendix).

As illustrated in Table 13 (refer Appendix), the overwriting attack demonstrates a minimal impact
on the rendering quality of IPR-NeRF (*i.e.* rendering quality only experiences a slight decrease).
However, the embedded watermark is completely compromised, resulting in a substantial reduction
and significant deterioration in the quality of the extracted watermark. As such, the extracted water-
mark is unusable for ownership protection. Conversely, the digital signature in IPR-NeRF remains
highly persistent in protecting the ownership, showcasing a BER of 0. In summary, we can con-
clude that this overwriting attack effectively replaces the embedded watermark without significantly
compromising the rendering quality of the protected NeRF model. Nevertheless, it is ineffective in
overwriting the digital signature, which remains resilient to this overwrite attack. Hence, the own-
ership claim of the protected NeRF model remains valid through the embedded digital signature.

## 5 DISCUSSION AND CONCLUSION

This paper introduces a complete and robust NeRF-IPR protection scheme in both black-box and
white-box scenarios. Comprehensive experimental results demonstrate its effectiveness in resist-
ing ambiguity and removal attacks on the embedded watermark while maintaining rendering per-
formance. However, it has limitations in computational power and black-box protection against
overwriting attacks when the attacker possesses detailed knowledge about the protected model. Fu-
ture research will focus on improving these aspects. This study offers significant value to NeRF
model developers and researchers, providing a way to protect their intellectual property and gain
a competitive advantage in the market, considering the substantial resources required for develop-
ing a high-performing NeRF model. Strengthening NeRF models against IPR violations has broad
societal benefits, including preventing plagiarism, ensuring a competitive edge in dynamic market
competitiveness, and the burden of wasteful lawsuit cases.

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

# A  APPENDIX

## A.1  OVERVIEW OF IPR-NERF OWNERSHIP VERIFICATION PROCESS

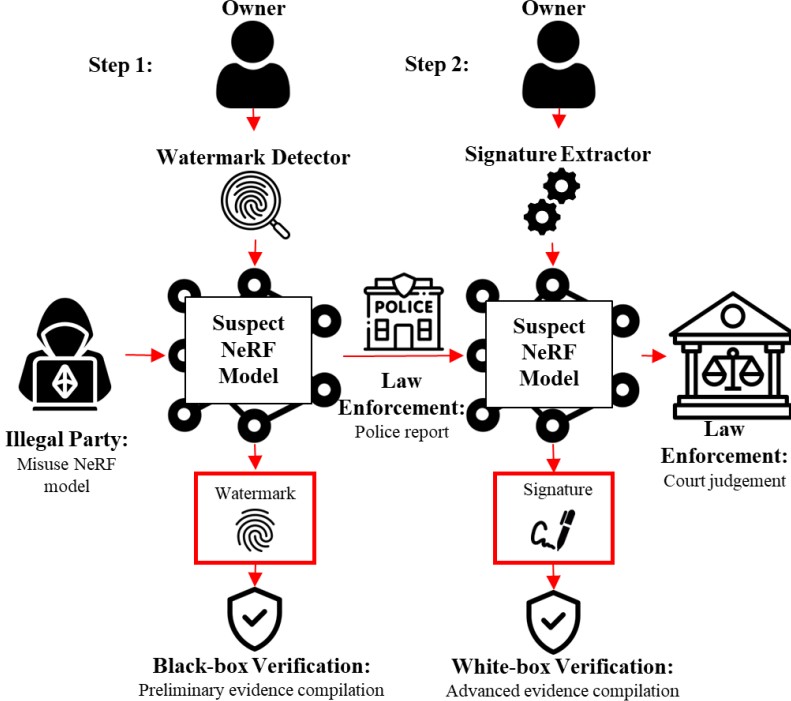

Figure 12: The ownership verification process of the proposed IPR-NeRF framework involves two key steps: black-box verification (Step 1) and white-box verification (Step 2). In step 1, a watermark detector is employed on the rendered image of the suspected NeRF model to extract the embedded watermark, forming the basis for preliminary evidence compilation. In step 2, the embedded signature within the model's weights is extracted to enable advanced evidence collection.

## A.2  PRELIMINARY RESULTS

| Method | Training Image | Embedded Watermark | Extracted Watermark |
|---|---|---|---|
| DeepStega Baluja (2020) | | | |
| HiDDen Zhu et al. (2018) | | | |

Table 4: Preliminary results of applying 2D steganography methods by watermarking the training images, resulting in an unrecoverable watermark during NeRF rendering.

| Model | BER↓ |
|---|---|
| NeRF with $wm$ | 0 |
| NeRF with $wm'$ | 0 |

Table 5: Preliminary result of applying Uchida et al. (2017) method to protect NeRF model with genuine watermark, $wm$ and forged watermark, $wm'$. The result indicate that the embedded watermark is able to be counterfeit easily, potentially leading to a situation of ambiguity.

### A.3 SSIM VERSUS VISIBILITY

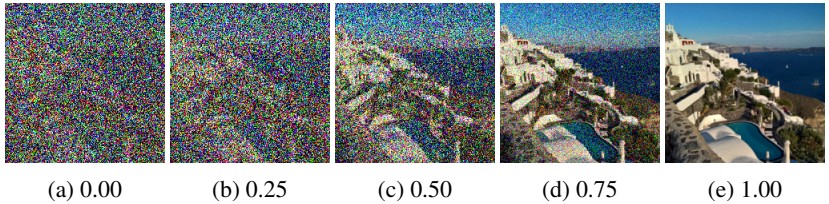

(a) 0.00      (b) 0.25      (c) 0.50      (d) 0.75      (e) 1.00

Figure 13: Varying perceived quality of the watermark image and the corresponding SSIM scores.

### A.4 ALGORITHM AND IMPLEMENTATIONS

---

**Algorithm 1** Training step of the two-optimization of IPR-NeRF

---

1: **Input:** Training images $I_T$, Watermark image $w$, Original NeRF $\theta_o$, Camera pose $P$, Signature $s$, Learning rate $\eta$
2: Optimize $\theta_o$ to converged
3: Load well-trained $\theta_o$ to $\theta_m$ as initial weights
4: initialise detector $d$
5: **for** all number of training iterations **do**
6:      Sample random $P$ to render, $x = \theta_m(P)$
7:      $\mathcal{L}_d \leftarrow \mathcal{L}_d(d, x, w)$ in Eq. 3
8:      $\mathcal{L}_{photometric} \leftarrow \mathcal{L}_{photometric}(P)$ in Eq. 7
9:      $\mathcal{L}_s \leftarrow \mathcal{L}_s(s)$ in Eq. 6
10:      $\mathcal{L} \leftarrow \mathcal{L}_{photometric} + \lambda_d \mathcal{L}_d + \mathcal{L}_s$ in Eq. 8
11:      Take gradient descent step on $\eta \cdot \nabla_\theta(\mathcal{L})$ and $\eta \cdot \nabla_d(\mathcal{L})$ to update $\theta_m$ and $d$ respectively
12: **end for**
13: **Output:** Optimized IPR-NeRF $\theta_m$ and diffusion-based watermark detector $d$

---

### A.5 FALSE POSITIVE DETECTION PREVENTION

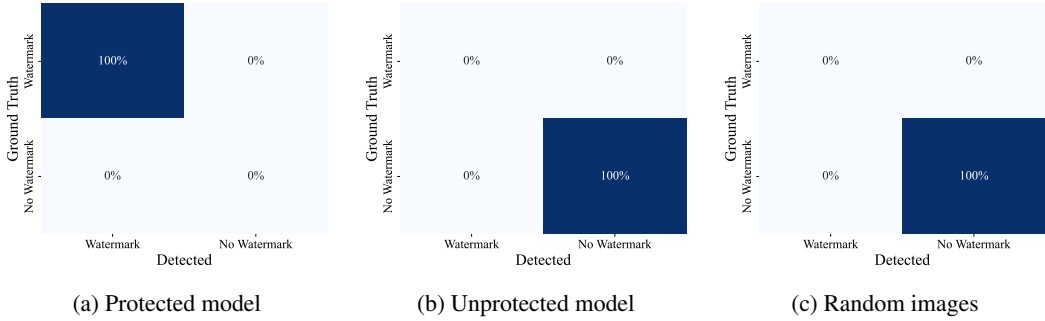

(a) Protected model      (b) Unprotected model      (c) Random images

Figure 14: Confusion matrix of the detected watermark from the rendered image of the protected model, rendered image of the unprotected model and random images based on the proposed watermark decision threshold of $w_{\text{SSIM}} > 0.75$ as discussed in Sec. 3.4.

The details of Fig. 5 are provided here. Let $x$ be the rendered views of IPR-NeRF, $y$ be the rendered views of an unprotected standard NeRF (same dataset and same scene as a protected one), and $z$ be any random images (*e.g.*, downloaded from the internet). Both the protected and unprotected NeRF models in this experiment employed scenes from the NeRF-Synthetic 360 dataset, including Lego, Chair, and Drum, as well as scenes from the LLFF-Forward dataset, namely Fern, Fortress, Room, and Flower. We compute the deterministic Gaussian noise through Eq. 4 to obtain $x_T$, $y_T$ and $z_T$. We then extract the watermark with Eq. 5 to obtain $\hat{w}_x$, $\hat{w}_y$ and $\hat{w}_z$. To quantitatively evaluate the false positive detection, we measure SSIM between the ground-truth watermark $w$ and the extracted

watermarks $\hat{w}_y/\hat{w}_z$. We find that the averaged SSIM is only $\approx 0.05 - 0.10$, which is essentially a noise image (see Fig. 13(a)). That is, we calculated the false positive rate (FPR) and false negative rate (FNR) for watermark extraction using the trained watermark detector ($w_{\text{SSIM}} > 0.75$), resulting in a value of 0 for each as shown in Fig. 14. In simpler terms, our trained detector exclusively extract watermarks from the deterministic forward distribution of the protected model. This capability arises because the detector has been specifically tailored to the deterministic forward distribution of the protected model through the proposed joint optimization process. Consequently, it may not effectively remove the deterministic Gaussian noises present in other random images, such as those generated from rendered views of an unprotected NeRF model. The quantitative and qualitative results of the recovered watermark is shown in Tab. 6 and Fig. 15 respectively.

To further understand why the false positives did not happen, we compute the histogram of 100 different views from each scene of $x_T/y_T$ and 100 random images of $z_T$. We then measure the maximum mean discrepancy (MMD) and Wasserstein distance (WD) between $x_T$ and $y_T$ and $z_T$. As illustrated in Fig. 5, the distribution of $y_T$ and $z_T$ is very different from $x_T$ with the MMD/WD discrepancy of 0.2979/2.77 and 0.8238/10.16 respectively. Note that only distribution of $x$ is close to G. This empirically proves that the detector can prevent a false positive detection.

|  | $w_{\text{SSIM}}$ |
| --- | --- |
| Protected model | 0.9526±0.0218 |
| Unprotected model | 0.0593±0.0037 |
| Random images | 0.0847±0.0185 |

Table 6: Quantitative result of the various recovered watermarks from the rendered image of the protected model, rendered image of the unprotected model and random images.

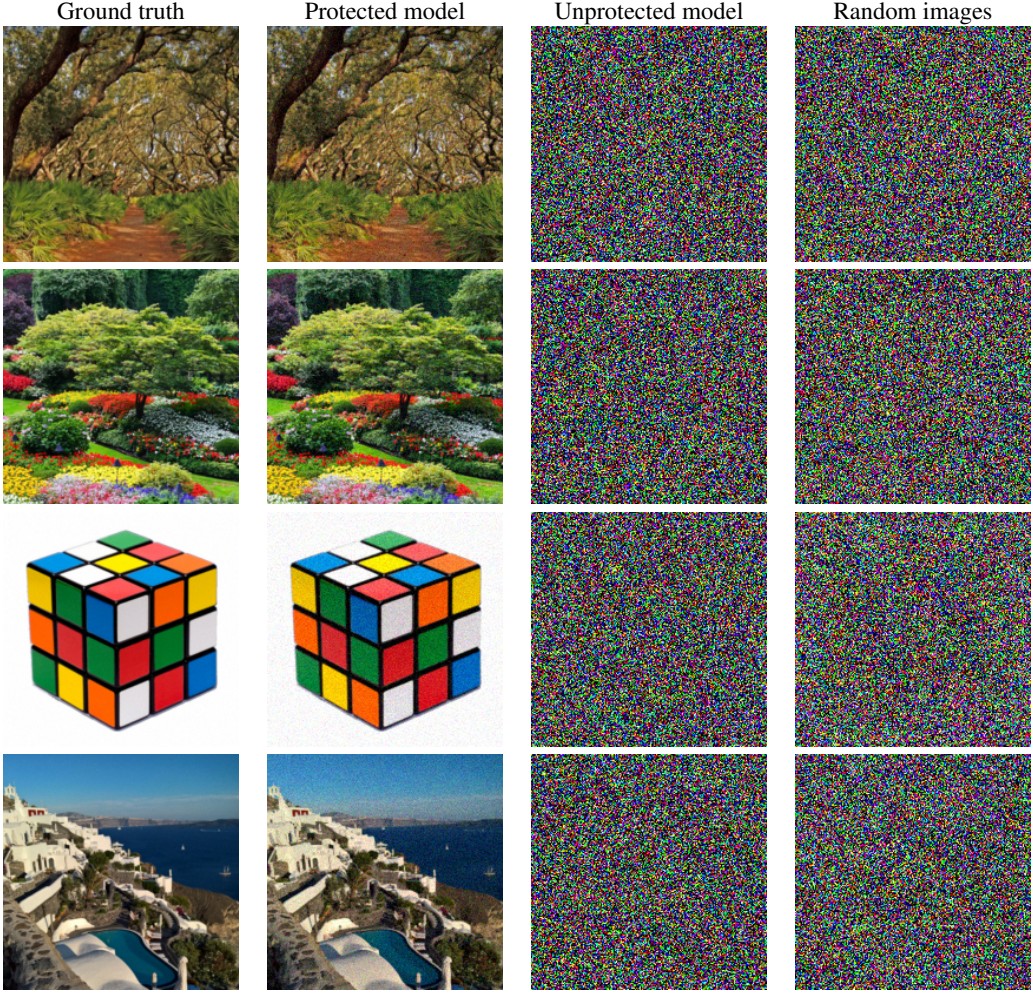

Figure 15: Qualitative result of the various recovered watermarks from the rendered image of the protected model, rendered image of the unprotected model and random images.

## A.6 Digital Signature Extraction

| **B** | | | **O** | | | **B** | | |
|---|---|---|---|---|---|---|---|---|
| $\gamma$ | bit | ASCII | $\gamma$ | bit | ASCII | $\gamma$ | bit | ASCII |
| -0.40 | 0 | | -0.67 | 0 | | -0.37 | 0 | |
| 0.31 | 1 | | 0.58 | 1 | | 0.31 | 1 | |
| -0.66 | 0 | | -0.39 | 0 | | -0.36 | 0 | |
| -0.68 | 0 | 66 | -0.31 | 0 | 79 | -0.34 | 0 | 66 |
| -0.35 | 0 | | 0.45 | 1 | | -0.63 | 0 | |
| -0.33 | 0 | | 0.33 | 1 | | -0.45 | 0 | |
| 0.72 | 1 | | 0.37 | 1 | | 0.74 | 1 | |
| -0.30 | 0 | | 0.65 | 1 | | -0.33 | 0 | |

Table 7: Example of the trained batch normalization weight $\gamma$ of IPR-NeRF when embedded digital signature, $S = \{BOB\}$.

## A.7 Fidelity Analysis

| | | NeRF (baseline) | IPR-NeRF (Ours) | StegaNeRF | CopyRNeRF |
|---|---|---|---|---|---|
| NeRF-Synthetic | PSNR ↑ | 31.21 ($\pm$0.28) | 31.05 ($\pm$0.46) | 31.14 ($\pm$0.42) | 30.93 ($\pm$0.63) |
| | SSIM ↑ | 0.9593 ($\pm$0.0048) | 0.9571 ($\pm$0.0062) | 0.9585 ($\pm$0.0027) | 0.9531 ($\pm$ 0.0074) |
| | LPIPS ↓ | 0.0528 ($\pm$0.0029) | 0.0573 ($\pm$0.0051) | 0.0537 ($\pm$0.0039) | 0.0584 ($\pm$0.0085) |
| LLFF-Forward | PSNR ↑ | 27.59 ($\pm$0.21) | 27.39 ($\pm$0.0032) | 27.42 ($\pm$ 0.0291) | 27.21 ($\pm$0.0081)- |
| | SSIM ↑ | 0.8293 ($\pm$0.0083) | 0.8237 ($\pm$0.0077) | 0.8277 ($\pm$0.0049) | 0.8224 ($\pm$0.0093) |
| | LPIPS ↓ | 0.1685 ($\pm$0.0025) | 0.1705 ($\pm$0.0042) | 0.1693 ($\pm$0.0053) | 0.1718 ($\pm$0.0032) |

Table 8: Fidelity: Comparison of the proposed IPR-NeRF rendering quality against StegaNeRF Li et al. (2023) and CopyRNeRF Luo et al. (2023) in different metrics.

## A.8 Rendering Results From Different Camera Poses

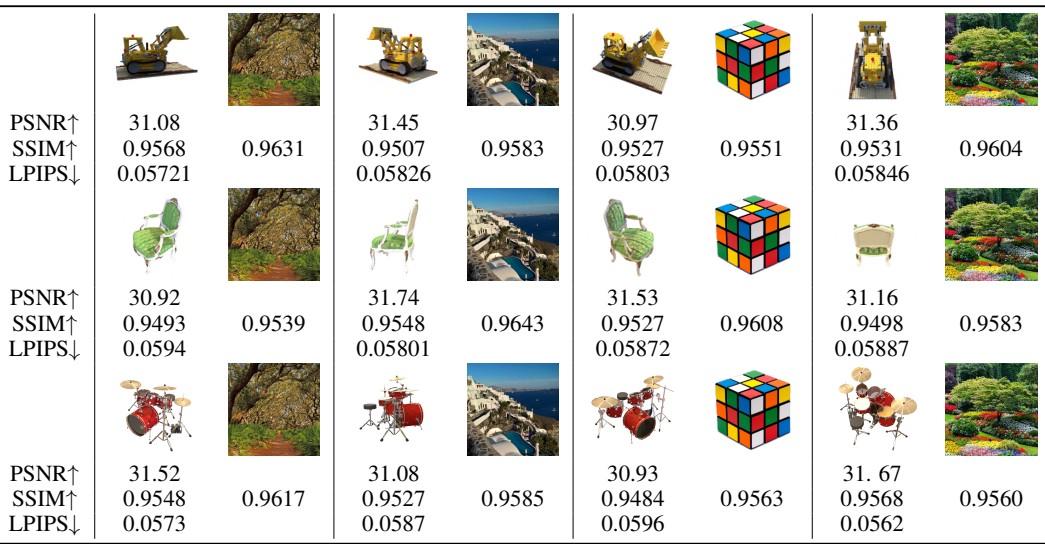

| | | | | | | | | |
|---|---|---|---|---|---|---|---|---|
| PSNR↑ | 31.08 | | 31.45 | | 30.97 | | 31.36 | |
| SSIM↑ | 0.9568 | 0.9631 | 0.9507 | 0.9583 | 0.9527 | 0.9551 | 0.9531 | 0.9604 |
| LPIPS↓ | 0.05721 | | 0.05826 | | 0.05803 | | 0.05846 | |
| PSNR↑ | 30.92 | | 31.74 | | 31.53 | | 31.16 | |
| SSIM↑ | 0.9493 | 0.9539 | 0.9548 | 0.9643 | 0.9527 | 0.9608 | 0.9498 | 0.9583 |
| LPIPS↓ | 0.0594 | | 0.05801 | | 0.05872 | | 0.05887 | |
| PSNR↑ | 31.52 | | 31.08 | | 30.93 | | 31. 67 | |
| SSIM↑ | 0.9548 | 0.9617 | 0.9527 | 0.9585 | 0.9484 | 0.9563 | 0.9568 | 0.9560 |
| LPIPS↓ | 0.0573 | | 0.0587 | | 0.0596 | | 0.0562 | |

Table 9: Quantitative and qualitative results from our proposed IPR-NeRF framework involve rendering with various embedded watermarks and camera poses, along with the corresponding extracted watermarks from the NeRF-Synthetic datasets. The rendering quality is assessed using PSNR, SSIM, and LPIPS metrics. Simultaneously, the quality of the extracted watermark is evaluated based on the SSIM metric.

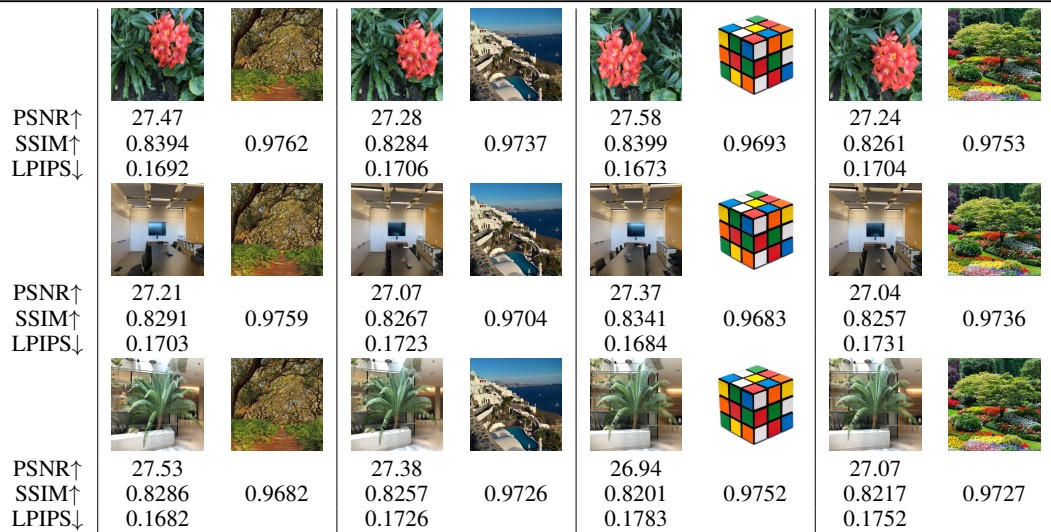

| | | | | | | | |
|---|---|---|---|---|---|---|---|
| PSNR↑ | 27.47 | | 27.28 | | 27.58 | | 27.24 | |
| SSIM↑ | 0.8394 | 0.9762 | 0.8284 | 0.9737 | 0.8399 | 0.9693 | 0.8261 | 0.9753 |
| LPIPS↓ | 0.1692 | | 0.1706 | | 0.1673 | | 0.1704 | |
| PSNR↑ | 27.21 | | 27.07 | | 27.37 | | 27.04 | |
| SSIM↑ | 0.8291 | 0.9759 | 0.8267 | 0.9704 | 0.8341 | 0.9683 | 0.8257 | 0.9736 |
| LPIPS↓ | 0.1703 | | 0.1723 | | 0.1684 | | 0.1731 | |
| PSNR↑ | 27.53 | | 27.38 | | 26.94 | | 27.07 | |
| SSIM↑ | 0.8286 | 0.9682 | 0.8257 | 0.9726 | 0.8201 | 0.9752 | 0.8217 | 0.9727 |
| LPIPS↓ | 0.1682 | | 0.1726 | | 0.1783 | | 0.1752 | |

Table 10: Quantitative and qualitative rendering results from our proposed IPR-NeRF framework involve rendering with various embedded watermarks and camera poses, along with the corresponding extracted watermarks from the LLFF-Forward datasets. The rendering quality is assessed using PSNR, SSIM, and LPIPS metrics. Simultaneously, the quality of the extracted watermark is evaluated based on the SSIM metric.

## A.9 EXTENDED RENDERING RESULTS ON DIFFERENT SCENES

| | NeRF | | IPR-NeRF w/o $s$ | | IPR-NeRF (Full) | |
|---|---|---|---|---|---|---|
| | Render | Watermark | Render | Watermark | Render | Watermark |
| **NeRF-Synthetic** | 0.9582 | N/A | 0.9551 | 0.9639 | 0.9573 | 0.9658 |
| | 0.9547 | N/A | 0.9521 | 0.9621 | 0.9538 | 0.9652 |
| **LLFF-Forward** | 0.8274 | N/A | 0.8235 | 0.9781 | 0.8253 | 0.9729 |
| | 0.8214 | N/A | 0.8193 | 0.9732 | 0.8208 | 0.9757 |

Table 11: The qualitative and quantitative results of the IPR-NeRF for different scenes in the two datasets: NeRF-Synthetic (mic and ship scenes) and LLFF-Forward (orchid and trex scenes). Accompanying each set of rendering results is the corresponding extracted watermark image. The performance is evaluated in SSIM.

## A.10 EXTENDED RENDERING RESULTS ON UNBOUND-360 DATASET

| | | NeRF | | IPR-NeRF | |
| --- | --- | --- | --- | --- | --- |
| | | Render | Watermark | Render | Watermark |
| **Garden** | PSNR↑
SSIM↑
LPIPS↓ | 22.38
0.5385
0.4417 | N/A | 22.16
0.5359
0.4483 | 0.9526 |
| **Kitchen** | PSNR↑
SSIM↑
LPIPS↓ | 25.52
0.7219
0.3789 | N/A | 25.27
0.7182
0.3994 | 0.9625 |

Table 12: The qualitative and quantitative rendering results of the NeRF and IPR-NeRF for garden and kitchen scenes in Unbound-360 dataset Barron et al. (2022). The rendering quality is assessed using PSNR, SSIM, and LPIPS metrics. Simultaneously, the quality of the extracted watermark is evaluated based on the SSIM metric.

## A.11 QUALITATIVE RESULTS ON MODEL PRUNING

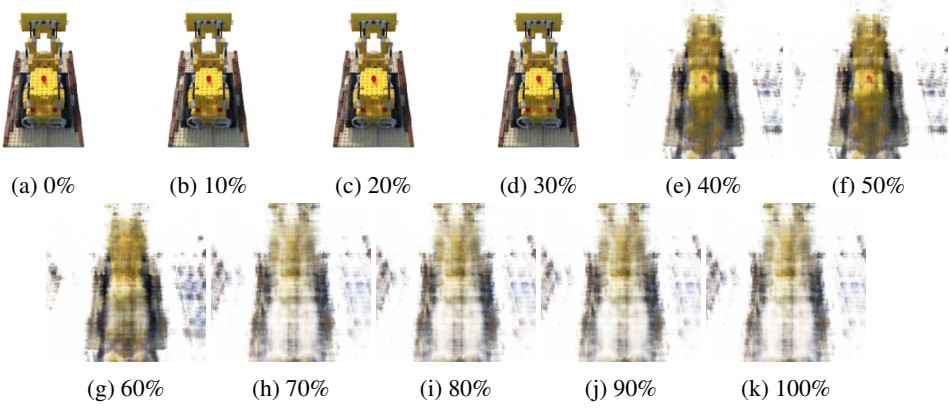

(a) 0%    (b) 10%    (c) 20%    (d) 30%    (e) 40%    (f) 50%

(g) 60%    (h) 70%    (i) 80%    (j) 90%    (k) 100%

Figure 16: Qualitative rendering results of different model pruning rates.

Although our approach successfully embeds a watermark, we observed that it became compromised when the model was pruned at rates exceeding 40%. This resulted in the watermark becoming unextractable, as demonstrated in Fig. 10 and Fig. 11. However, as depicted in Fig. 16, when the model is pruned at rates exceeding 40%, not only does the embedded watermark become unextractable, but the rendering quality of the NeRF model also significantly deteriorates. Consequently, it renders the model unusable and diminishes its commercial value.

## A.12 QUANTITATIVE RESULTS ON REMOVAL ATTACKS

| | | IPR-NeRF (Ours) | StegaNeRF | CopyRNeRF |
|---|---|---|---|---|
| NeRF-Synthetic | SSIM ↑ | 0.9308 | 0.9273 | 0.9375 |
| | $w_{\text{SSIM}}$ ↑ | 0.3517 | 0.2819 | 0.3216 |
| | BER ↓ | 0 | ✗ | ✗ |
| LLFF-Forward | SSIM ↑ | 0.8174 | 0.7947 | 0.8061 |
| | $w_{\text{SSIM}}$ ↑ | 0.4173 | 0.3602 | 0.3916 |
| | BER ↓ | 0 | ✗ | ✗ |

Table 13: Comparatively analysis of our approach against StegaNeRF and CopyRNeRF on overwrite attack.

## A.13 TIME COMPLEXITY

As shown in Table 14, our approach requires significantly more training time than the original NeRF framework (Mildenhall et al., 2020). This is due to our proposed two-stage optimization process for embedding watermarks into rendering. However, it's important to note that our approach does not impact inference time. The inference time of our approach remains the same as that of the original NeRF framework. Given that the inference process is a frequent operation for users. In contrast, the watermark embedding process is an infrequent task, we confidently affirm that our approach is exceptionally well-suited for real-world deployment. It robustly safeguards intellectual property rights and ownership of the NeRF model.

| | Relative Time | |
|---|---|---|
| | Training | Inference |
| NeRF | 1.00 | 1.00 |
| IPR-NeRF w/o $s$ | 1.58 | 1.00 |
| IPR-NeRF (Full) | 1.59 | 1.00 |

Table 14: Comparative analysis of the training and inference times for the proposed IPR-NeRFframework in comparison to the original NeRF framework. The values presented are relative to those of the original NeRF framework.

## A.14 ABLATION STUDY OF COEFFICIENT $\lambda$

In Section 3.1, we introduced $\lambda$ to scale the watermark detection term $\mathcal{L}_D$. This balance is pivotal for the rendering quality of the protected NeRF model and the watermark image. We performed an ablation study on $\lambda$, and the results are in Table 15.

| $\lambda_d$ | 0.1 | 0.5 | 1.0 | 2.5 | 5.0 |
|---|---|---|---|---|---|
| SSIM ↑ | 0.9784 | 0.9682 | 0.9541 | 0.9359 | 0.8715 |
| $w_{\text{SSIM}}$ ↑ | 0.9217 | 0.9437 | 0.9671 | 0.9751 | 0.9781 |

Table 15: Impact of $\lambda_d$ on IPR-NeRFrendering performance and quality of the extracted watermark image measured in SSIM.

From the data presented in Table 15, we can discern a clear relationship: when $\lambda$ is set to a low value, the rendering quality of the protected NeRF model is notably high, but this comes at the expense of the quality of the extracted watermark image, which is generally lower and vice-versa. In summary, our analysis reveals a tradeoff between the rendering quality of the protected NeRF model and the quality of the extracted watermark image. Notably, our findings demonstrate that an optimal balance between these two factors is achieved when $\lambda$ is set to 1.0. At this value, the quality of the extracted watermark image remains relatively good, without adversely impacting the performance of the protected NeRF's rendering quality.

## A.15 LIMITATION

Although our proposed approach of watermarking (black-box) on rendered scenes, coupled with embedding a digital signature (white-box) into the NeRF model's weights, demonstrates superior performance in protecting against unauthorized usage, as demonstrated in Section 4, it comes with some inevitable limitations.

Firstly, the current approach for watermarking the NeRF model demands substantial computational power, as it necessitates a gradient descent update on the weights of the watermark NeRF model during the proposed two-stage optimization process. Secondly, our suggested black-box protection scheme is susceptible to overwriting attacks. In the worst-case scenario, where the attacker possesses comprehensive knowledge about the model, the embedded watermark for black-box verification can be entirely removed, as detailed in Section 4.5. This limitation imposes restrictions on open-sourcing the protected NeRF model, as disclosing training steps is essential to prevent others from compromising it. We anticipate that future research will address this issue by further reducing the required computational resources, allowing users to protect their NeRF models more efficiently and ensuring comprehensive defense against overwriting attacks.

