# OpenReview forum: "IPR-NeRF: Ownership Verification Meets Neural Radiance Field"
_ICLR.cc/2024/Conference — Submitted to ICLR 2024_

### Official Review · Reviewer_qCQu · 2023-10-18

**Soundness:** 3 good
**Presentation:** 1 poor
**Contribution:** 2 fair
**Rating:** 6
**Confidence:** 3

**Summary:**

Since intellectual property rights (IPR) matter for NeRFs, this paper argues that the proposed method has high fidelity, effectiveness, and robustness against adversarial activities and situations. For the black-box protection, they effectively incorporate a diffusion method to generate a watermark from novel-view generations of NeRFs. For the white-box protection, they propose to use the signs of normalization layers in the NeRF model. They experimentally validate their method in aspects of fidelity, ownership verification with black/white-box schema, resilience against ambiguity attacks, and robustness against removal attacks, including model pruning and overwriting.

**Strengths:**

- The extensive validation confirms various aspects of watermarking and leaving and extracting signature methods.

- Robustness toward image degradation and forged signature seems to be promising.

**Weaknesses:**

- W1. Novelty and effectiveness. The Black-box watermark extraction method is a straightforward way to apply diffusion models. However, it relies on a catastrophic performance on unseen samples, only empirically validated on samplings (Sec 4.3). The below question Q2 raises an issue of false positive detection.

- W2. Clarity on the white-box method. The author did not clearly state where are the normalization layers in the NeRF model. As raised in question Q1, the readers cannot pinpoint the normalization layers in the original NeRF model, so reproducibility is unclear.

**Questions:**

- Q1. Which ones are the normalization layers in the NeRF model? The author states that they used the original implementation in Mildenhall et al. (2020), but their implementation does not have any normalization layers.

- Q2. In Fig 5, a substantial portion of the protected model (red) and random images (blue) are overlapping. How do you say that they are significantly different?

- In Sec 3, Para 2, by initiate -> by initiating

- Appendix A.5, to qualitatively -> to quantitatively

---

> ### Author Response · Authors · 2023-11-16
>
> Dear Reviewer,
>
> We appreciate your thoughtful review of our NeRF model implementation and thank you for raising pertinent questions. We would like to address your concerns and provide further clarification on the normalization layers and other aspects of our manuscript.
>
> (a)Clarification on Normalization Layers:
> - We sincerely apologize for the oversight. Yes, it is correct that the original NeRF model [1] did not have normalization layers. In our work, we modified the NeRF model by appending normalization layers to all Multi-Layer Perceptron (MLP) layers. This modification aimed to embed a digital signature, optimized through sign loss objectives, thereby enhancing Intellectual Property Rights (IPR) protection in a white-box scenario.
> - Our experiments (Table 2) demonstrated that the introduction of these normalization layers did not adversely impact the rendering performance. We will include this description in the revised version of our manuscript to ensure reproducibility.
>
> [1] https://arxiv.org/pdf/2003.08934.pdf
>
> (b)Response to Figure 5 Distribution:
> - Although a substantial portion of the distribution of the protected model and the random images overlaps, there is still a difference of 0.2979 and 2.77 in the computed average maximum mean discrepancy and average Wasserstein distance across all scenes. The details of each scene are as follows:
>
> |  NeRF-Synthetic| Lego   | Chair  | Drum   | Average    |
> |-----------------------|--------|--------|--------|------------|
> | Maximum mean discrepancy| 0.2876         | 0.3427 | 0.3343 | 0.3215 |
> | Wasserstein distance   | 2.3769         | 2.9267 | 2.8434 | 2.7157 |
>
> | LLFF-Forward   | Fern   | Fortress | Room   | Flower | Average|
> |-----------------------|--------|----------|--------|--------|--------|
> | Maximum mean discrepancy| 0.2714         | 0.3372 | 0.3136   | 0.1984 | 0.2802 |0.2802|
> | Wasserstein distance   | 2.2140         | 2.8717 | 2.6364   | 3.5205 | 2.8107 |2.8107|
>
>
> - The trained diffusion-based watermark detector can only extract a watermark from the overfitted distribution of the protected model. As a result, it is unable to extract the watermark from the distribution of random images and the unprotected model in which the extracted image is essentially a noise image similar to watermark image extracted by 2D steganography method (see Figure 1 of the main paper).
> - This is because the trained diffusion-based watermark detector was exclusively and strongly overfitted to the distribution of the protected model, effectively preventing false positive detections. We apologize for any confusion caused by our writing and will address this issue by refining our manuscript in the revised version.
>
> (c)Apology for Grammatical Mistakes:
> - Turning to the matter of grammatical errors, we extend our sincere apologies for any oversights. We have revised our manuscript, focusing specifically on rectifying grammatical mistakes, including those highlighted by your insightful feedback.
>
> We value your constructive feedback and assure you that our revised manuscript will comprehensively address and clarify the points you have raised. Your input is invaluable, and we are committed to ensuring that our work meets the rigorous standards of the conference.

---

> ### Comment · Reviewer_qCQu · 2023-11-20
> **Feedbacks**
>
> Thanks for the clarification on the implementation details of the normalization layers. It is a *crucial* point in your work.
>
> You'd better show the false positive rate (FPR) and false negative rate (FNR) using the proposed decision boundary ($w_\text{SSIM}$ > 0.75), rather than the average mean discrepancy or Wasserstein distance. The choice of evaluation metric makes the judgment of the proposed method difficult to assess its effectiveness. The authors just mentioned "preventing a false positive detection" but didn't say how much to prevent that.

---

> ### Author Response · Authors · 2023-11-21
>
> Clarification on the Implementation Details of the Normalization Layers: Thank you, this is to confirm that we have included the normalization layer implementation details in the revised manuscript to ensure reproducibility.
>
> False Positive Rate (FPR) and False Negative Rate (FNR) using the Proposed Decision Boundary: As suggested, we show the false positive rate (FPR) and false negative rate (FNR), along with the true positive rate (TPR) and true negative rate (TNR) for completeness, employing the proposed decision boundary ($w_\text{SSIM} > 0.75$) as below,
>
> |                 | TPR  |  TNR   | FPR | FNR |
> |-----------------|------|------|----|-----|
> | Protected Model | 1 |  0   |  0 | 0   |
> | Unprotected Model| 1  |  0   |  0 | 0   |
> | Random Images    | 1  |  0   |  0 | 0   |
>
> It indicates that the trained watermark detector will only effectively detect watermarks from the protected model. We will include this result in the paper.
>
> Note that, the True Positive (TP) for the protected model represents the correct detection of the watermark in the watermarked scene. On the other hand, the TP for both the unprotected model and random images indicates the correct non-detection of the watermark in unmarked scenes.

---

> > ### Comment · Reviewer_qCQu · 2023-11-22
> > **A question on the comment**
> >
> > The number of samples, to be reported along with the results, is omitted.
> >
> > Anyway, I'm still confused about interpreting the results. As you mentioned in the paper: "In Fig. 5, we observe that the deterministically computed noise distribution (through Eq. 4) of rendered views from unprotected models or any random images ...", and Fig. 5 shows the significant overlaps among random, unprotected model, and protected model; how do we get the *perfect* performance in the detection of watermark?

---

> > > ### Author Response · Authors · 2023-11-22
> > >
> > > Dear Reviewer,
> > >
> > > Clarification on Number of Samples:
> > >
> > > Sorry for the confusion. The detailed of the experiment was included in the revised manuscript in Section A.5. To conduct this evaluation, both the protected and unprotected NeRF models utilized scenes from the NeRF-Synthetic 360 dataset, encompassing Lego, Chair, Drum, as well as scenes from the LLFF-Forward dataset, specifically Fern, Fortress, Room, and Flower. In total, we utilized 100 different views from each of these scenes and 100 random images in the experiment.
> > >
> > > Clarification on Detection of Watermark:
> > >
> > > For better clarity, the noise image distribution (depicted in Figure 5) was calculated deterministically using our proposed diffusion-based detector through Equation 4 (forward sampling). To recover the watermark, our detector will require a deterministic reverse sampling, as presented in Equation 5. This step entails computation based on the noise image generated during the preceding forward sampling by the detector. Therefore, in our case, to extract a watermark, it must rely on the deterministic forward distribution of the protected model. This implies that the extraction of the watermark is dependent on the deterministic forward distribution of the protected model, to which the detector has been overfitted via our proposed joint optimization process.
> > >
> > > Hence, despite the deterministic forward distribution of the unprotected model and random images overlapping with that of the protected model, the trained detector is incapable of extracting a watermark from them. The resulting image extracted from the rendered images of the unprotected model and random images is merely an unusable noise image with a low structural similarity index ($w_\text{SSIM}$), as illustrated in the table below and Figure 15 (Section A.5) of our newly revised manuscript.
> > >
> > > In summary, our trained detector exclusively extract watermarks from the deterministic forward distribution of the protected model. This capability arises because the detector has been specifically tailored to the deterministic forward distribution of the protected model through the proposed joint optimization process. Consequently, it may not effectively remove the deterministic Gaussian noises present in other random images, such as those generated from rendered views of an unprotected NeRF model.
> > >
> > > |                 | $w_\text{SSIM}$  |
> > > |-----------------|------|
> > > | Protected Model | 0.9526$\pm$0.0218 |
> > > | Unprotected Model| 0.0593$\pm$0.0037  |
> > > | Random Images    | 0.0847$\pm$0.0185  |
> > >
> > > To confirm, we run the experiment 5 times, and reported the mean/standard deviation as above. As shown, with $w_\text{SSIM}$ set as > 0.75, the trained watermark detector successfully detects the watermark only in the rendering of the protected model, while unable to extract the watermark in the rendering of the unprotected model and random images.
> > >
> > > Hope this clarifies your confusion.

---

> > > > ### Comment · Reviewer_qCQu · 2023-11-23
> > > >
> > > > Thank you for your clarification and endeavor to promptly respond in an open discussion. Hope you will update your manuscript accordingly.
> > > >
> > > > Since the authors faithfully responded via rebuttal, I raised my score one step leaning toward acceptance.

---

> ### Author Response · Authors · 2023-11-23
> **Thank you for your engagement.. And updated draft**
>
> We are delighted to have addressed your concerns and would like to express our gratitude for your recognition of our revised version. Your thoughtful and constructive feedback has been integral to making these improvements possible.
>
> We have **updated** the Section 3.1 (black-box) and Section 4.3 (False positive detection). Specifically, we’ve mentioned in the updated Section 3.1:
> - This work was inspired by (Ho et al., 2020) to propose a diffusion-based method to learn the detector *d*. This is because the nature of diffusion models to gradually add noise to the image provides natural robustness against various forms of image degradation attacks as the main objective of diffusion models is denoising. Thanks to the powerful denoising ability of diffusion models, our idea is to first convert the views *x* into Gaussian noises, then we denoise from these Gaussian noises into the watermark *w*. As a result, we can seamlessly translate between *x* and *w* through *d*.
>
> Then in Section 3.2 (white-box), we added
> - To achieve this, normalization layers were appended to the MLP implementation.
>
> Also, we have updated the **Section 4.3**, as well as **Appendix A.5** with experiment details:
> - Both the protected and unprotected NeRF models in this experiment employed scenes from the NeRF-Synthetic 360 dataset, including Lego,  Chair, and Drum, as well as scenes from the LLFF-Forward dataset, namely Fern, Fortress, Room,
> and Flower.  To further understand why the false positives did not happen, we compute the histogram of 100 different views from each scene of protected/unprotected and 100 random images.
> - A new confusion matrix (Figure 14) and Figure 15 (Qualitative result of the various recovered watermarks from the rendered image of the protected model, rendered image of the unprotected model and random images) are added to further clarify the False Positive Detection Prevention.
>
> In the appendix, new experiment using **Unbound-360 Dataset** is also included in **Appendix A.10** to clarify the impact of dataset selection on ownership verification.
> - We conducted an experiment using the 'Garden' and 'Kitchen' scenes from the Unbound-360 dataset. We found that the quality of the recovered watermark still maintained a high standard with an SSIM of 0.95 overall (Kitchen: watermark SSIM 0.9625; Garden: watermark SSIM 0.9526).
>
> Finally, the font in all figures has been adjusted to ensure better clarity and comprehension.

---

### Official Review · Reviewer_Cayy · 2023-10-30

**Soundness:** 2 fair
**Presentation:** 2 fair
**Contribution:** 1 poor
**Rating:** 5
**Confidence:** 2

**Summary:**

This paper suggests an intellectual property protection framework for the NeRF model in both black-box and white-box settings. IPR-NeRF used a diffusion-based model to embed and extract watermark images for black-box ownership protection. In addition, it adopted a technique from DeepIPR method to embed a textual signature into the NeRF for a white-box protection scheme.

**Strengths:**

1. To the best of my knowledge, It first proposes to embed both watermarking and textual signature simultaneously.
2. Using diffusion models seems to be quite effective compared to the previously suggested simple decoder-based methods.
3. Various experiments to show the robustness of the proposed scheme make the paper more convincing.

**Weaknesses:**

1. My major concern is that It lacks the original technical contribution. It is interesting to see that the diffusion models (DDIM) are quite effective in this task. However, I think it may not be a sufficient contribution to be a full conference paper.
2. For textual embedding, the authors adopted the technique from DeepIPR work into neural fields. It is also useful information to the community, but I still think the direct application to the well-defined neural network architecture would not be a sufficient technical contribution.
3. I wonder if the DDIM method can generate high-fidelity images. The comparison to the previous method, e.g., staganerf would be great.

**Questions:**

Questions are embedded in the weakness section.

---

> ### Author Response · Authors · 2023-11-15
>
> Dear Reviewer,
>
> Thank you for your thoughtful comments and concerns regarding the technical contribution of our work. We appreciate the opportunity to clarify the key points you raised.
>
> (a) Technical Contribution to Black-box Protection:
> - We propose diffusion models as the foundation for a watermark detector, enabling the extraction of embedded watermarks within the rendered images of NeRF models. Our methodology has demonstrated noteworthy resilience against noise, a point substantiated in our research. More importantly, our work stands as a pioneering effort in uncovering IPR protection strategies for NeRF models, a pursuit of paramount significance for the 3D vision community. The escalating commercial use of NeRF, particularly in applications like the metaverse, underscores the urgency for a robust NeRF IPR protection framework.
>
> - While our black-box protection methodology hinges on diffusion models, it's crucial to highlight the non-trivial nature of jointly optimizing watermark injection into the rendered scenes of NeRF (Eq 8). This optimization task becomes even more intricate when striving for a high watermark recovery rate. Consequently, the primary contribution of our work lies in pioneering the application of diffusion models for black-box NeRF protection. We demonstrate the feasibility of the joint optimization process (Eq 8) for embedding watermarks into rendered scenes and extracting embedded watermarks using diffusion models (Eq 4 and Eq 5), a valuable insight for the 3D community. In essence, our work contributes to the evolving landscape of NeRF IPR protection. Also, it serves as a testament to the viability and applicability of diffusion models in the joint optimization process. We believe these advancements are crucial steps toward a more secure and resilient NeRF IPR protection framework.
>
> (b) Technical Contribution to White-box Protection:
> - We acknowledge the existence of DeepIPR [3] but contend that its technical contribution might be deemed insufficient. Upon reviewing previous works on IPR protection for NeRF (e.g. StegaNeRF [1] and CopyRNeRF [2]), we observed a predominant focus on black-box protection strategies. This reliance on black-box protection becomes a point of vulnerability. In scenarios where black-box protection is compromised or when black-box ownership verification is not verified, the owner of the protected NeRF model may find it challenging to assert ownership in cases of unauthorized misuse. Given the substantial resources required to train a high-performance NeRF model, this ambiguity surrounding ownership could lead to significant financial losses.
>
> - Considering these factors, we have proactively integrated the DeepIPR technique, widely recognized as a robust white-box protection framework against various IPR attacks in the neural network IP protection research field. By doing so, this work offers complete IPR protection for the NeRF model. The comprehensive IPR protection framework proposed in our work not only addresses potential compromises in any one protection aspect but also aims to elevate the overall security of NeRF models. Moreover, the intent in applying DeepIPR's white-box protection extends beyond the immediate scope of our work. We aspire to raise awareness among future researchers in the IPR domain of NeRF models. We recommend and demonstrate that both white-box and black-box protection strategies be considered to ensure the safety and effectiveness of NeRF model protection. In this way, we envision future work on NeRF IPR protection to be comprehensive in its approach, covering both white-box and black-box dimensions.
>
> [1] https://arxiv.org/abs/2212.01602
>                [2] https://arxiv.org/abs/2307.11526
> [3] https://www.computer.org/csdl/journal/tp/2022/10/09454280/1uqBfxOEjny
>
> (c) Fidelity Analysis:
> - Thanks! We appreciate the suggestion. And Yes! Our proposed method does not affect the fidelity of the rendered images. We have included a comprehensive fidelity comparison between our proposed method and the existing methods, namely StegaNeRF[1] and CopyRNeRF [2] in our revised manuscript, as well as below.
>
> |                     | NeRF   | IPR-NeRF(Ours) | StegaNeRF | CopyRNeRF |
> |------------------|--------|----------------|-----------|-----------|
> |**NeRF Synthetic**|
> | **PSNR** | 31.21  |31.05  |31.14| 30.93|
> | **SSIM** | 0.9593 |0.9571  |0.9585    | 0.9531    |
> | **LPIPS**| 0.0528 |0.0573 | 0.0537    | 0.0584    |
> |**LLFF-Forward**|
> | **PSNR**    | 27.59  |27.39 | 27.42     | 27.21     |
> | **SSIM**    | 0.8293 |0.8237|0.8277    | 0.8224    |
> | **LPIPS**   | 0.1685 |0.1705|0.1693    | 0.1718    |
>
> We hope these clarifications demonstrate the significance of our contributions and address your concerns. We appreciate your valuable feedback.

---

> > ### Comment · Area_Chair_jDk7 · 2023-11-22
> > **Discussion with authors**
> >
> > Dear Reviewer CAyy:
> >
> > Thanks for the review. The authors have uploaded their responses to your comments, please check if the rebuttal address your concerns and if you have further questions/comments to discuss with the authors. If the authors have addressed your concerns, please adjust your rating accordingly.
> >
> > AC

---

> > > ### Comment · Reviewer_Cayy · 2023-11-22
> > > **Response and final rating**
> > >
> > > Thanks for your thorough response and additional experiments, I do appreciate it. I carefully read your response, and I can raise my score to 5. However, I still think the original contribution might not be sufficient, and if I have to decide on one side, I am leaning towards not accepting the current draft.

---

> ### Author Response · Authors · 2023-11-23
> **Thank you for your review.**
>
> We are glad that our responses helped and would like to thank you for raising the score of our paper.

---

> ### Author Response · Authors · 2023-11-23
> **Updated Draft**
>
> We have **updated** the Section 3.1 (black-box) and Section 4.3 (False positive detection). Specifically, we’ve mentioned in the updated Section 3.1:
> - This work was inspired by (Ho et al., 2020) to propose a diffusion-based method to learn the detector *d*. This is because the nature of diffusion models to gradually add noise to the image provides natural robustness against various forms of image degradation attacks as the main objective of diffusion models is denoising. Thanks to the powerful denoising ability of diffusion models, our idea is to first convert the views *x* into Gaussian noises, then we denoise from these Gaussian noises into the watermark *w*. As a result, we can seamlessly translate between *x* and *w* through *d*.
>
> Then in Section 3.2 (white-box), we added
> - To achieve this, normalization layers were appended to the MLP implementation.
>
> Also, we have updated the **Section 4.3**, as well as **Appendix A.5** with experiment details:
> - Both the protected and unprotected NeRF models in this experiment employed scenes from the NeRF-Synthetic 360 dataset, including Lego,  Chair, and Drum, as well as scenes from the LLFF-Forward dataset, namely Fern, Fortress, Room,
> and Flower.  To further understand why the false positives did not happen, we compute the histogram of 100 different views from each scene of protected/unprotected and 100 random images.
> - A new confusion matrix (Figure 14) and Figure 15 (Qualitative result of the various recovered watermarks from the rendered image of the protected model, rendered image of the unprotected model and random images) are added to further clarify the False Positive Detection Prevention.
>
> In the appendix, new experiment using **Unbound-360 Dataset** is also included in **Appendix A.10** to clarify the impact of dataset selection on ownership verification.
> - We conducted an experiment using the 'Garden' and 'Kitchen' scenes from the Unbound-360 dataset. We found that the quality of the recovered watermark still maintained a high standard with an SSIM of 0.95 overall (Kitchen: watermark SSIM 0.9625; Garden: watermark SSIM 0.9526).
>
> Finally, the font in all figures has been adjusted to ensure better clarity and comprehension.

---

### Official Review · Reviewer_hWgR · 2023-11-03

**Soundness:** 3 good
**Presentation:** 3 good
**Contribution:** 2 fair
**Rating:** 6
**Confidence:** 2

**Summary:**

This paper presents an intellectual property protection framework for NeRF models in black-box (i.e., w/o access to model weights) and white-box (i.e., w/ access to model weights). Specifically, the authors adopt a diffusion-based model for embedding and extracting watermark in rendered images by NeRF, in the black-box setting. A digital signature is embedded into NeRF model weights for the white-box setting. The experiments on LLFF-forward and NeRF-Synthetic datasets verify that the protection framework can identity the protected NeRF without degrading the rendering quality.

**Strengths:**

1. Well-motivated: Since NeRF-based 3D reconstruction is more and more easy-to-use, people are sharing their NeRF models on web. Thus, it is worth exploring how to alleviate copying, re-distributing, or misusing those models.

2. Impressive Results: The baseline works can protect NeRF models but degrade the rendering quality obviously. This work nearly does not affect the rendering quality.

3. Easy-to-follow draft: the draft is well-written and the figures are easy-to-understand.

**Weaknesses:**

1. Scalability to explicit NeRF representations: To accelerate NeRF inference and rendering, multiple works [1,2,3] have proposed to use explicit representations (e.g., grid, mesh, and point cloud) instead of MLP as the NeRF representations. Modifying the weights of explicit NeRF representations seems to have a larger effect on the rendering quality as compared to implicit representations because it can be regarded as changing the location/color of the grid/mesh/point cloud. Thus, it is not sure whether the proposed protection framework can still maintain the rendering quality of those explicit NeRF representations.

2. Insufficient details in "False Positive Detection Prevention": In Sec. A.5, the dataset used in this experiment is not mentioned. Does Fig. 5 show the averaged histogram for different scenes in a specific dataset or the histogram for a specific scene? The unprotected standard NeRF is claimed to be trained in the same dataset as a protected one. Is it trained on the same scene or a different scene but in the same dataset?

3. Will the dataset affect the selection of ownership verification? A threshold of 0.75 is set for the black-box ownership verification, and the author claimed that 0.75 is selected "as the visibility is still evident." However, it is uncertain whether different datasets have different optimal thresholds. For example, scenes in Unbound-360 [4] or ARKitScene [5] is more close to the real-world applications [6,7]. Will the optimal threshold still be 0.75 on those datasets?

[1] https://creiser.github.io/merf/

[2] https://mobile-nerf.github.io/

[3] https://repo-sam.inria.fr/fungraph/3d-gaussian-splatting/

[4] https://jonbarron.info/mipnerf360/

[5] https://github.com/apple/ARKitScenes

[6] https://poly.cam/gaussian-splatting

[7] https://lumalabs.ai/interactive-scenes

**Questions:**

See weakness

---

> ### Author Response · Authors · 2023-11-16
>
> Dear Reviewer,
>
> Thank you for your valuable comments and thoughtful inquiries concerning our work. We appreciate the opportunity to respond to your concerns and are committed to providing additional clarity in our revised submission.
>
> (a)Scalability to explicit NeRF representations:
> - Although our proposed method was designed for implicit NeRF representations, it can be *directly* applied to explicit NeRF representations. This is because our black-box diffusion-based detector does not depend on the architecture of NeRF models but rather on the rendered images. Additionally, our white-box protection scheme only adds/modifies the normalization layer into the MLP layers.
> We will conduct a further investigation on directly applying our proposed methods to explicit NeRF representations in our future work.
>
> (b)Clarification on "False Positive Detection Prevention" Section A.5 in Manuscript:
> -  We apologize for any confusion caused by the oversight in our writing. For clarification, both the protected and unprotected NeRF models employed scenes from the NeRF-Synthetic 360 dataset, including Lego, Chair, and Drum, as well as scenes from the LLFF-Forward dataset, namely Fern, Fortress, Room, and Flower. We are committed to incorporating this important information into our revised manuscript to enhance the clarity and quality of our work.
>
> (c)Clarification on the Impact of Dataset Selection on Ownership Verification:
> - For clarification, different datasets  would not require to have different optimal thresholds. We are sorry for the misunderstanding. This section primarily centers around the crucial quality of extracted watermarks for asserting ownership, for which we chosen the threshold = 0.75 (SSIM). This choice is made empirically, aligning with Figure 13, where the watermark is still distinctly discernible for ownership claim. Additionally, our experimental findings (refer to Table 2) consistently demonstrate that the quality of the extracted watermark consistently surpasses 0.95 (SSIM) across diverse scenes within the NeRF-Synthetic and LLFF-Forward datasets.
> - Additionally, we conducted an experiment using the 'Garden' and 'Kitchen' scenes from the Unbound-360 dataset [1], as suggested. We found that the quality of the recovered watermark still maintained a high standard with an SSIM of 0.95 overall (Kitchen: watermark SSIM 0.9625; Garden: watermark SSIM 0.9526). We have included these additional experimental results in our revised manuscript for better clarification.
>
> [1] https://jonbarron.info/mipnerf360/
>
> Your constructive feedback is invaluable, and we assure you that we will address these issues promptly in our revised manuscript to enhance the clarity and quality of our work. We appreciate the opportunity to address these concerns in our revised manuscript.

---

> ### Author Response · Authors · 2023-11-23
> **Update Draft**
>
> We have **updated** the Section 3.1 (black-box) and Section 4.3 (False positive detection). Specifically, we’ve mentioned in the updated Section 3.1:
> - This work was inspired by (Ho et al., 2020) to propose a diffusion-based method to learn the detector *d*. This is because the nature of diffusion models to gradually add noise to the image provides natural robustness against various forms of image degradation attacks as the main objective of diffusion models is denoising. Thanks to the powerful denoising ability of diffusion models, our idea is to first convert the views *x* into Gaussian noises, then we denoise from these Gaussian noises into the watermark *w*. As a result, we can seamlessly translate between *x* and *w* through *d*.
>
> Then in Section 3.2 (white-box), we added
> - To achieve this, normalization layers were appended to the MLP implementation.
>
> Also, we have updated the **Section 4.3**, as well as **Appendix A.5** with experiment details:
> - Both the protected and unprotected NeRF models in this experiment employed scenes from the NeRF-Synthetic 360 dataset, including Lego,  Chair, and Drum, as well as scenes from the LLFF-Forward dataset, namely Fern, Fortress, Room,
> and Flower.  To further understand why the false positives did not happen, we compute the histogram of 100 different views from each scene of protected/unprotected and 100 random images.
> - A new confusion matrix (Figure 14) and Figure 15 (Qualitative result of the various recovered watermarks from the rendered image of the protected model, rendered image of the unprotected model and random images) are added to further clarify the False Positive Detection Prevention.
>
> In the appendix, new experiment using **Unbound-360 Dataset** is also included in **Appendix A.10** to clarify the impact of dataset selection on ownership verification.
> - We conducted an experiment using the 'Garden' and 'Kitchen' scenes from the Unbound-360 dataset. We found that the quality of the recovered watermark still maintained a high standard with an SSIM of 0.95 overall (Kitchen: watermark SSIM 0.9625; Garden: watermark SSIM 0.9526).
>
> Finally, the font in all figures has been adjusted to ensure better clarity and comprehension.

---

### Official Review · Reviewer_eU8D · 2023-11-10

**Soundness:** 3 good
**Presentation:** 3 good
**Contribution:** 2 fair
**Rating:** 6
**Confidence:** 4

**Summary:**

This paper presents IPR-NeRF, an intellectual property protection framework for NeRF models. It offers protection in both black-box and white-box settings. In the black-box approach, a watermark is embedded and extracted using a diffusion-based method. In the white-box scenario, a digital signature is incorporated into the NeRF model's weights using a sign loss objective. The experiments show that IPR-NeRF maintains rendering quality while being robust against ambiguity and removal attacks, providing a solution to safeguard NeRF models from unauthorized use and distribution.

**Strengths:**

- I think the paper investigates an important and interesting topic in the 3D vision community.
- The paper is well-written and easy to follow.
- The comprehensive experimental results significantly demonstrate the benefit of the proposed method.

**Weaknesses:**

- The motivation for using the diffusion model to learn black-box protection is unclear. It would be great if the authors could provide more elaboration.

**Questions:**

Major:
- Is there any simple alternative solution for black-box protection? If so, could the author provide some comparisons?
- Just out of curiosity, is there any reason that StageNeRF is significantly vulnerable to Gaussian noise?

Minor:
- Enlarging the font in the figures could be helpful.

---

> ### Author Response · Authors · 2023-11-15
>
> Dear Reviewer,
>
> Thank you for your insightful comments and queries regarding our work on applying the diffusion model for black-box protection in NeRF. We appreciate the opportunity to address your concerns and provide additional clarification.
>
> (a) Elaboration on Applying the Diffusion Model:
> - Our initial findings using simple solutions such as DeepStega [1] and HiDDen [2] as well as more advanced methods such as StegaNeRF [3] demonstrated a few issues: (i) Ineffective watermark extraction on the rendered scene [1,2] due to the embedded watermark being smooth out during rendering (please refer to Figure 1 in the main paper); and (ii) The extraction of watermarks from the rendered scene is not robust to the introduction of noise [3], as it was trained under noise-free conditions (please refer to Figure 6 and Figure 7 in the main paper)
>
> - As a result, we propose to use diffusion model as the foundation for our black-box approach. First, the nature of diffusion models to gradually add noise to the image provides natural robustness against various forms of image degradation attacks as the main objective of diffusion models is denoising (please refer to Figure 6 and Figure 7 in main paper). This resilience is a result of the inherent characteristics of the diffusion model. Second, the process of watermark embedding has minimal impact on the rendering quality of the NeRF models (please refer to Table 2 in main paper). Third, it can retrieve embedded watermarks within the rendered images of NeRF models (please refer to Table 2 in main paper).
>
> - We sincerely apologize for the lack of an explanation of our motivation for applying the diffusion model as the black-box protection, we will add the above description to our revised manuscript.
>
> (b) Simple Alternative Solution for Black-Box Protection of NeRF:
> - Yes, some examples of simple alternative solutions are DeepStega [1] and HiDDen [2], which involve directly watermarking the training images before model training. However, as illustrated in Figure 1 of the main paper, the embedded watermark could not be extracted as it was smoothed out during training and rendering. Hence, these simple alternative methods are deemed not suitable for black-box protection of NeRF models.
>
> [1] https://proceedings.neurips.cc/paper_files/paper/2017/file/838e8afb1ca34354ac209f53d90c3a43-Paper.pdf
> [2] https://dl.acm.org/doi/10.1007/978-3-030-01267-0_40
> [3] https://arxiv.org/abs/2212.01602
>
> (c) Vulnerability of StegaNeRF to Gaussian Noise:
> - The vulnerability of StegaNeRF [3] to Gaussian noise is due to the ideal conditions of the watermark detector's training procedure, which does not consider ``noise'' as a form of attack. Therefore, the watermark detector in StegaNeRF is implemented as a simple autoencoder-based detector. Consequently, when noise is added to the rendered scene as a form of removal attack, the features of the rendered image are compromised, causing the embedded watermark unable to be extracted effectively.
>
> (d) Addressing Font Clarity in Figures:
> - We sincerely apologize for the oversight regarding the unclear font in the figures, including legends and axis labels. To enhance the reader's viewing experience, we will enlarge the font in these elements to ensure better clarity and comprehension in the revised manuscript.
>
> We hope these responses adequately address your concerns, and we are committed to implementing the necessary improvements in our revised manuscript.

---

> > ### Comment · Area_Chair_jDk7 · 2023-11-22
> > **Discussion with authors**
> >
> > Dear Reviewer eU8D:
> >
> > Thanks for the review. The authors have uploaded their responses to your comments, please check if the rebuttal address your concerns and if you have further questions/comments to discuss with the authors. If the authors have addressed your concerns, please adjust your rating accordingly.
> >
> > AC

---

> ### Author Response · Authors · 2023-11-23
> **Updated Draft**
>
> We have **updated** the Section 3.1 (black-box) and Section 4.3 (False positive detection). Specifically, we’ve mentioned in the updated Section 3.1:
> - This work was inspired by (Ho et al., 2020) to propose a diffusion-based method to learn the detector *d*. This is because the nature of diffusion models to gradually add noise to the image provides natural robustness against various forms of image degradation attacks as the main objective of diffusion models is denoising. Thanks to the powerful denoising ability of diffusion models, our idea is to first convert the views *x* into Gaussian noises, then we denoise from these Gaussian noises into the watermark *w*. As a result, we can seamlessly translate between *x* and *w* through *d*.
>
> Then in Section 3.2 (white-box), we added
> - To achieve this, normalization layers were appended to the MLP implementation.
>
> Also, we have updated the **Section 4.3**, as well as **Appendix A.5** with experiment details:
> - Both the protected and unprotected NeRF models in this experiment employed scenes from the NeRF-Synthetic 360 dataset, including Lego,  Chair, and Drum, as well as scenes from the LLFF-Forward dataset, namely Fern, Fortress, Room,
> and Flower.  To further understand why the false positives did not happen, we compute the histogram of 100 different views from each scene of protected/unprotected and 100 random images.
> - A new confusion matrix (Figure 14) and Figure 15 (Qualitative result of the various recovered watermarks from the rendered image of the protected model, rendered image of the unprotected model and random images) are added to further clarify the False Positive Detection Prevention.
>
> In the appendix, new experiment using **Unbound-360 Dataset** is also included in **Appendix A.10** to clarify the impact of dataset selection on ownership verification.
> - We conducted an experiment using the 'Garden' and 'Kitchen' scenes from the Unbound-360 dataset. We found that the quality of the recovered watermark still maintained a high standard with an SSIM of 0.95 overall (Kitchen: watermark SSIM 0.9625; Garden: watermark SSIM 0.9526).
>
> Finally, the font in all figures has been adjusted to ensure better clarity and comprehension.

---

### Author Response · Authors · 2023-11-22
**Rebuttal Summary**

We would like to express our gratitude to all the reviewers for their dedicated efforts and time spent reviewing our paper. We deeply appreciate the valuable and detailed feedback provided. We are particularly thankful for the positive comments:
- affirming the motivation behind our work;
- this paper is the first to introduce a complete intellectual property rights (IPR) protection for NeRF. We are delighted that reviewers recognize its significance, anticipating numerous new prospects for the 3D vision community; and
- comprehensive experimental results significantly demonstrate the benefit of the proposed method.

While there are some minor issues in our paper, mostly related to description-related aspects, we have independently addressed and clarified these issues with each reviewer. Also, we have taken great care to resolve them in the revised version.

---

### Meta-Review · Area_Chair_jDk7 · 2023-12-04

**Metareview:**

This paper investigates how to protect IP of the NeRF models in both black-box and white-box settings. It porposes an IPR-NeRF framework to achieve the goal. In the black-box setting, a watermark is embedded and extracted using a diffusion-based method. In the white-box setting, a digital signature is incorporated into the NeRF model's weights using a sign loss objective.
The experiments on LLFF-forward and NeRF-Synthetic datasets verify that the protection framework can identity the protected NeRF without degrading the rendering quality.

Strengths:
+ The proposed method provides comprehensive IPR protection of NeRF.
+ Using diffusion model is effective.
+ Extensive experimental results demonstrate its robustness toward image degradation and forged signature.
+ The paper is well written and easy to follow.

Weaknesses:
- The novelty and technical contribution is limited: The Black-box watermark extraction method is a straightforward way to apply diffusion models; textual signature embedding is adopted from DeepIPR work.
- It lacks scalability to explicit NeRF representations.
- There are insufficient details in "False Positive Detection Prevention".
- The motivation of using diffusion model for black-box setting is unclear.
- It is not clear whether the dataset affects the selection of ownership verification.

**Justification For Why Not Higher Score:**

This is a borderline paper. Reviewers acknowledge that this investigation is worthwhile. However, there are shared concerns on the motivation, novelty, scalability, and limited technical contribution of the paper.
The authors address some of these, providing additional clarification and revised paper, but these were not enough to sway reviewers. I think this paper is not ready for publication at the current stage, and give the authors more time to improve the paper.

**Justification For Why Not Lower Score:**

N/A

---

### Decision · Program_Chairs · 2024-01-16

Reject